# Lead-Free Bi_3.15_Nd_0.85_Ti_3_O_12_ Nanoplates Filler-Elastomeric Polymer Composite Films for Flexible Piezoelectric Energy Harvesting

**DOI:** 10.3390/mi11110966

**Published:** 2020-10-28

**Authors:** Wancheng Qin, Peng Zhou, Yajun Qi, Tianjin Zhang

**Affiliations:** Ministry of Education Key Laboratory for Green Preparation and Application of Functional Materials, Hubei Provincial Key Laboratory of Polymers, School of Materials Science and Engineering, Hubei University, Wuhan 430062, China; wanchengqin@gmail.com (W.Q.); p_zhou@outlook.com (P.Z.)

**Keywords:** flexible piezoelectric energy harvesting, Bi_3.15_Nd_0.85_Ti_3_O_12_ single crystalline, nanoplates, lead-free

## Abstract

Nowadays, wearable and flexible nanogenerators are of great importance for portable personal electronics. A flexible piezoelectric energy harvester (f-PEH) based on Bi_3.15_Nd_0.85_Ti_3_O_12_ single crystalline nanoplates (BNdT NPs) and polydimethylsiloxane (PDMS) elastomeric polymer was fabricated, and high piezoelectric energy harvesting performance was achieved. The piezoelectric output performance is highly dependent on the mass ratio of the BNdT NPs in the PDMS matrix. The as-prepared f-PEH with 12.5 wt% BNdT NPs presents the highest output voltage of 10 V, a peak-peak short-circuit current of 1 μA, and a power of 1.92 μW under tapping mode of 6.5 N at 2.7 Hz, which can light up four commercial light emitting diodes without the energy storage process. The f-PEHs can be used to harvest daily life energy and generate a voltage of 2–6 V in harvesting the mechanical energy of mouse clicking or foot stepping. These results demonstrate the potential application of the lead-free BNdT NPs based f-PEHs in powering wearable electronics

## 1. Introduction

With the rapid development of flexible and wearable electronics and the diffusion of sensor networks for the Internet of Things [1,2], the desire to replace batteries is the major force driving advances in energy harvesting technologies. Until now, various energy harvesting techniques have been successful demonstrated, including scavenging and converting ambient energy to electric energy by pyroelectric [3], piezoelectric [4] and triboelectric [5] effects. Of these methods, piezoelectric energy harvesting, which converts the ambient mechanical energy into useful electricity through piezoelectric transduction, has become the main research focus due to its high mechanoelectrical conversion efficiency, high stability, easy implementation and miniaturization [6,7,8]. Many piezoelectric materials, such as zinc oxide (ZnO) [9], gallium nitride (GaN) [10], lead zirconate titanate (PbZr_x_Ti_1-x_O_3_ (PZT)) [11], and poly(vinylidene fluoride) (PVDF)-based copolymers [12,13,14], have been used for creating piezoelectric energy harvesters (PEH), showing high energy conversion efficiency and high output voltage [15]. Integrating mechanical flexibility into energy harvesting devices extends application of the PEH towards the flexible and wearable electronics. The flexible PEHs (f-PEHs) fabricated with composites of inorganic piezoelectric nanostructures dispersed in polymer matrix exhibit high electrical output and high flexibility [16,17,18]. Various piezoelectric nanostructures, such as nanoparticles [19,20,21,22], nanowires [23,24,25,26] nanoflowers [27], nanofibers [28], and nanocubes [29] etc., have been used as fillers to disperse into polydimethylsiloxane (PDMS) [30] or polyvinylidene fluoride (PVDF) polymers [31], etc. It has also been found that the morphology of the fillers has an important influence on the output performance of the f-PEHs [22,29,32,33,34], because the high dimensional piezoelectric fillers enhance the continuity of fillers and also the transferring efficiency of stress from the matrix to the active inclusions [27], which thus improves the output performance of the PEHs [19]. Zhang et al. [16] reported in a groundbreaking work that a voltage of 65 V and a current of 75 nA were obtained by filling PDMS composite material with interconnected PZT foam. Gang et al. [27] obtained an even higher output open-circuit voltage of 260 V and a high short-circuit current of 50 μA in barium titanium trioxide (BaTiO_3_) flower filled PDMS composite PEHs. Furthermore, the size of the piezoelectric fillers should significantly influence the piezoelectric output performance of the f-PEHs since the piezoelectric coupling coefficients depend on the size of the nanostructures [22,35]. Additionally, the flexible composite nanogenerators based on nanoplates tend to have better piezoelectric performance because of the unidirectional orientation of the exposed plate surface [36]. As far as we know, however, research on the use of nanoplates/polymer composites in the fabrication of f-PEHs is not sufficient. Since the unique nanoplate structure is easy to distribute uniformly in the polymer matrix to form high spatial continuity, and the hard nanoplates can also serve as stress concentration points for generating a larger local deformation, it is expected to enhance the electrical output performance of f-PEHs with the addition of the nanoplate structure.

Aurivillius ferroelectrics, or bismuth-layered perovskite ferroelectrics, have attracted great attention due to their high Curie temperature and ferroelectric polarization, as well as large anisotropy in electromechanical coupling factors [37]. They exhibit excellent potential applications in nonvolatile ferroelectric random-access memory and high temperature piezoelectric sensor devices [38,39]. Specifically, bismuth titanate (Bi_4_Ti_3_O_12_) and lanthanide doped Bi_4_Ti_3_O_12_, which are prototypes of layer-structured ferroelectrics with high *T*_c_ (~675 °C) and large ferroelectric polarization (~50 µC/cm^2^) [40], are considered as potential candidates in the application of lead-free high-temperature piezoelectrics [41].

In this work, eco-friendly lead-free Bi_3.15_Nd_0.85_Ti_3_O_12_ (BNdT) nanoplates were fabricated and used as the filler in the PDMS matrix to form BNdT nanoplates (NPs)/PDMS composite film. The BNdT NPs/PDMS based f-PEH shows a piezoelectric output voltage of 10 V, a current of 1 μA and a power of 1.92 μW. It can also light up four commercial light-emitting diodes without the energy storage process. Moreover, the f-PEH is capable of harvesting energies with body movement, such as foot stepping, as well as clicking the mouse. The lead-free f-PEHs are environmentally friendly and can be used as a power source for flexible and wearable electronic devices. Furthermore, the proposed approach is simple, cost-effective, and applicable to fabricate large-scale high-performance f-PEHs.

## 2. Experimental Details

BNdT NPs were synthesized by a hydrothermal method. Appropriate amounts of bismuth nitrate pentahydrate (Bi(NO_3_)_3_•5H_2_O), bismuth nitrate pentahydrate (Nd(NO_3_)_3_) and bismuth nitrate pentahydrate (Ti(OC_4_H_9_)_4_) with a molar ratio of 3.15:0.85:3 were dissolved in 6 mL 2-methoxyethanolto to prepare the BNdT precursor with an concentration of 0.1 M. Then, 3 M bismuth nitrate pentahydrate (NaOH) solution used as the mineralizer. After stirring for 15 min, 0.2 g of poly-(ethylene glycol) with a molecular weight of 2000 was added into the solution, stirring for another 30 min. The mixture was then transferred into a Teflon-lined autoclave, which was sealed and heated up to 200 °C for 24 h after the autoclave was filled with deionized water up to 80% of its total volume. As the hydrothermal reaction finished, the reactor was naturally cooled to room temperature. The product was collected following the sequence of centrifugal separation, deionized water and ethanol washing, and then dried at 80 °C in air for 6 h. The microstructure and crystalline phase of the as-prepared BNdT products were examined by X-ray diffraction (XRD, D8 Advance Bruker, Germany), and transmission electron microscopy (TEM) (Titan G20, FEI, Hillsboro, OR, USA).

The PDMS (Sylgard 184, Dow Corning Corp., Auburn, MI, USA) polymer was prepared by mixing the PDMS precursor and cross linker at a weight ratio of 10:1, following which BNdT NPs were added into the PDMS solution and magnetically stirred to form a uniform mixture. The BNdT NPs with concentrations of 10, 12.5, 15 and 17.5 wt% (*w*/*v*) were added into the PDMS solution and stirred for 4 h. The mixture solution was spin-coated on a glass plate at 700 rpm for 30 s and cured at 80 °C for 1 h in a vacuum drying oven to obtain composite films. The dry composite films were peeled off from the glass plate and attached with Pt-coated polyimide (PI) substrate on both sides to form the f-PEHs.

A home-made tapping machine for applying an external force with a fixed reciprocating speed was used. A digital force gauge was used to measure the dynamic pressing force applied to the BNdT NPs/PDMS f-PEHs (Aipu Metrology Instrument Co., Ltd., Zhejiang, China). The open-circuit voltages of the f-PEHs were measured with a digital storage oscilloscope (TBS1072B, Tektronix, Beaverton, OR, USA) equipped with a passive probe (TPP0101, Tektronix, Beaverton, OR, USA). The short-circuit current was measured by an electrometer (2450, Keithley, Beaverton, OR, USA). Before electrical measurements, the as-prepared f-PEHs were poled under an electric field of 53.8 kV/cm for 10 min.

## 3. Results and Discussion

Figure 1a shows a *θ*−2*θ* scan X-ray diffraction (XRD) pattern of the synthesized BNdT NPs. All the diffraction peaks can be indexed according to the joint committee on powder diffraction standards (JCPDS) card no. 36-1486, showing that the as-prepared nanoplates are of layered-perovskite structure. No secondary phase was detected. The TEM image in the inset of Figure 1a shows that the as-prepared BNdT nanoplates exhibit a plate-like structure with a size in the range of 100–500 nm. Figure 1b shows a high-magnification TEM image of BNdT NPs. Two-dimensional lattice fringes were measured to be 0.271 and 0.273 nm in spacing, respectively, corresponding to the *d* values of (020) and (200) plane for BNdT, respectively. The inset of Figure 1b is the selected area electron diffraction (SAED) pattern of an individual NP. The clear diffraction spot array indicates the single crystalline nature of the nanoplate (NP), and it can be indexed as the (001) zone axis diffraction pattern, confirming the layered-perovskite structure.

Figure 2a illustrates the structure of BNdT NPs/PDMS based f-PEH. The lead-free BNdT NPs were well-dispersed in the PDMS matrix. Platinum (Pt) and PI were used as the electrode and sealing material in the power generating device, respectively. Figure 2b,c shows the flat and curved pictures of the device, indicating the flexible nature of the f-PEH. The active area of f-PEH is 3.5 cm × 2 cm. The cross-sectional SEM images of the BNdT NPs/PDMS composite film is shown in Figure 2d, from which a dense film with a thickness of ~130 µm is revealed. It is also clear that the piezoelectric BNdT NPs are randomly well-dispersed in the PDMS matrix, as indicated by arrows in the enlarged SEM image in Figure 2e.

The piezoelectric output of the BNdT NPs/PDMS f-PEHs were measured on a purpose-built platform in the repetitive tapping mode. Figure 3a–d shows the output performance of BNdT NPs/PDMS f-PEHs with various BNdT NP mass fractions. The generated voltage and current were collected using a digital storage oscilloscope and the repeating frequency was set as 2.7 Hz. All f-PEHs with various BNdT NP contents can generate voltages and exhibit content dependence. The generated voltages range from 4.2 to 10 V with increasing the BNdT NP mass fraction from 10 to 17.5 wt%. It is noted that the maximum output voltage ~10 V appeared in the f-PEHs with 12.5 wt% BNdT NPs. One may note that with the mass fraction varying from 10 to 17.5 wt%, the output voltage increases first and then decreases, with the maximum value reached at the mass fraction of 12.5 wt%. The piezoelectricity of the composites is enhanced by increasing the addition of inorganic BNdT NPs, which results in the increase in output voltage. On the other hand, further increasing the inorganic fillers will deteriorate the flexibility of the composites and the efficiency of strain transfer, leading to the decrease in the generated voltage [27].

The generated current was also measured, as shown in Figure 4a,b. The peak-peak short circuit current is about 1 μA. Switching-polarity tests were carried out to ensure that the generated signal is induced by piezoelectric effect. Negative output currents were observed when the device was reversely connected, as shown in Figure 4b. The output currents of the reversely connected device exhibit the same order of magnitude of values as that of the normally connected device, confirming the output signal is arisen from the piezoelectric phenomenon.

The output instantaneous power, voltage and current signals at various external load resistances, *R,* were also investigated, as shown in Figure 5. As *R* changed from 1 to 200 MΩ, the output voltage increased from 0.1 to 8 V, while the value of the maximum output current *I* decreased from 1.2 to 0.04 µA. The power is calculated by *p = I^2^R*, where *I* and *R* are the current and the load resistance, respectively. The calculated power *p* is presented in Figure 5b, where a maximum value of 1.92 µW is yielded at *R* = 1 MΩ, corresponding to a power density of 6.1 mW/m^2^.

We used COMSOL Multiphysics to simulate the stress and output electric potential of the f-PEH under applied force, as illustrated in Figure 6a,b. One can find that the effective stress is mainly extended to the piezoelectric NPs, and thus enhances the voltage output of the device due to the piezoelectric properties of the BNdT NPs. As shown in Figure 6a, by applying an external force of 6.5 N, a positive voltage (3 V) and a negative voltage (−2 V) are observed at the upper and lower edges of the BNdT NPs/PDMS composite, respectively. The simulation results further confirm the output voltage is arisen from the piezoelectricity of the BNdT NPs.

The generated voltage from the 12.5 wt% BNdT NPs/PDMS f-PEH can be rectified with the help of a rectifier bridge. The inset of Figure 7a shows the equivalent circuit diagram of the device. The generated power can be stored directly in a commercial capacitor, as presented in Figure 7a. A 47 μF capacitor had been charged to 0.5 V in 600 s by the 12.5 wt% BNdT NPs/PDMS f-PEH under periodic tapping of 6.5 N at a frequency of 2.7 Hz. Only positive signals were observed, as illustrated in Figure 7b.

The as-fabricated f-PEHs can be employed to harvest various types of ambient mechanical energy, such as foot stepping and mouse clicking. The 12.5 wt% BNdT NPs/PDMS f-PEH generates stable and continuous electrical voltage of about 2–4 V by foot stepping and mouse clicking, as shown in Figure 8. Furthermore, four commercial light emitting diodes (LEDs) are directly lit up by the 12.5 wt% BNdT NPs/PDMS f-PEH under repetitive foot stepping without any charge storage. These results demonstrate that the lead-free BNdT NPs/PDMS f-PEHs are applicable in harvesting various ambient energies.

## 4. Conclusions

In summary, lead-free f-PEHs based on BNdT NPs/PDMS composites were prepared and excellent energy harvesting performance was revealed. A maximum output voltage of 10 V, a peak-peak short-circuit current of about 1 μA, and a power density of 6.1 mW/m^2^ were produced in BNdT NPs/PDMS f-PEH with 12.5 wt% BNdT NPs under tapping mode. It can harvest ambient mechanical energy efficiently and can light up four LEDs without using storage capacitors. Our results indicate that the lead-free flexible generator has great potential in practical application for wearable electronic equipment.

## Figures and Tables

**Figure 1 micromachines-11-00966-f001:**
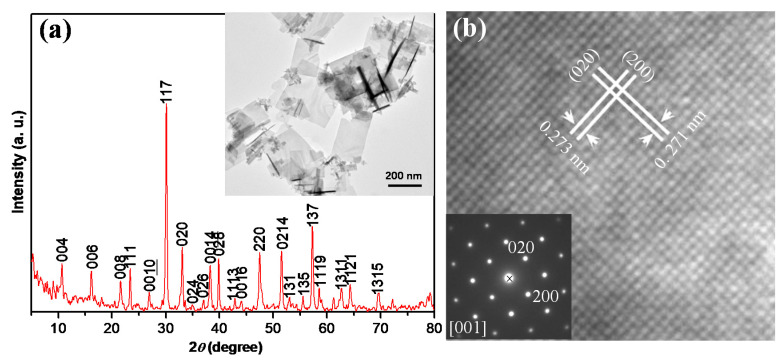
(**a**) XRD pattern of the as-synthesized Bi_3.15_Nd_0.85_Ti_3_O_12_ single crystalline nanoplates (BNdT NPs). Inset is the TEM image of the BNdT NPs, (**b**) High resolution transmission electron microscope (HRTEM) image of a BNdT NP. Inset is the selected area electron diffraction (SAED) pattern from the same nanoplate (NP).

**Figure 2 micromachines-11-00966-f002:**
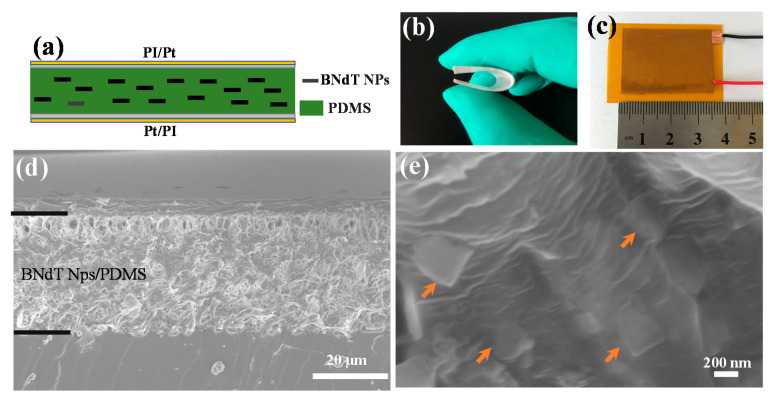
(**a**) Schematic diagram of the BNdT NPs/polydimethylsiloxane (PDMS) composites energy harvester; (**b**) and (**c**) photographs of the flexible BNdT NPs/PDMS composites piezoelectric energy harvester (PEH), (**d**) and (**e**) SEM images of BNdT NPs/PDMS composites. The arrows in (**e**) indicate the BNdT NPs.

**Figure 3 micromachines-11-00966-f003:**
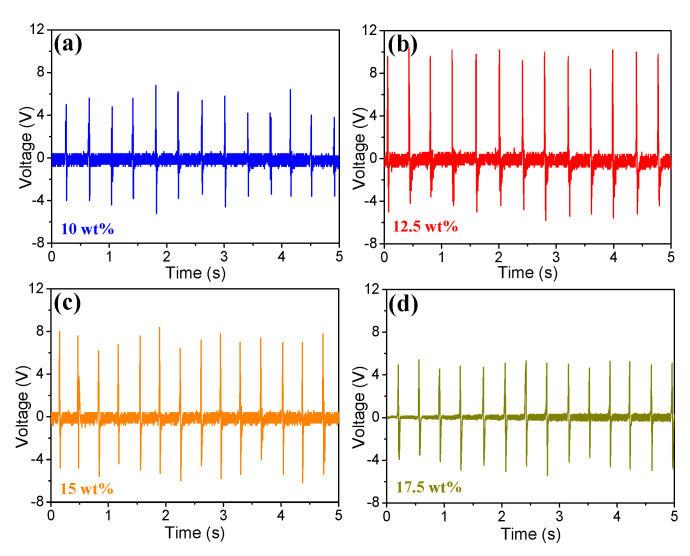
Output voltage signals of the flexible piezoelectric energy harvesters (f-PEHs) with various BNdT NP contents under repetitive tapping.

**Figure 4 micromachines-11-00966-f004:**
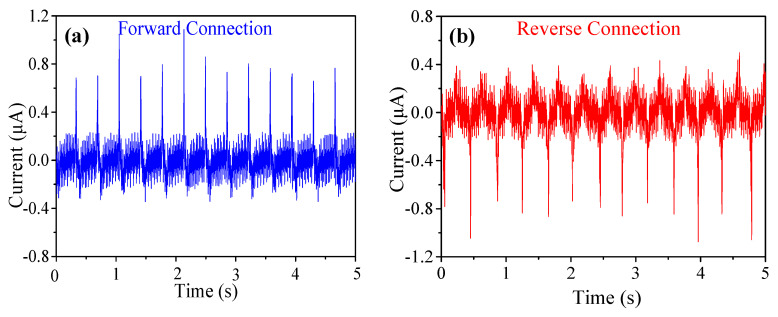
Short-circuit current output signals by repetitive tapping of 12.5 wt% BNdT NPs/PDMS f-PEH with (**a**) forward and (**b**) reverse connections with measuring instrument.

**Figure 5 micromachines-11-00966-f005:**
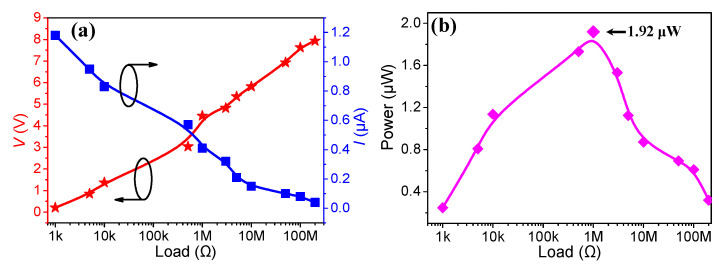
Variation of (**a**) output voltage *V* and current *I* and (**b**) the corresponding output power *p* under various external resistances.

**Figure 6 micromachines-11-00966-f006:**
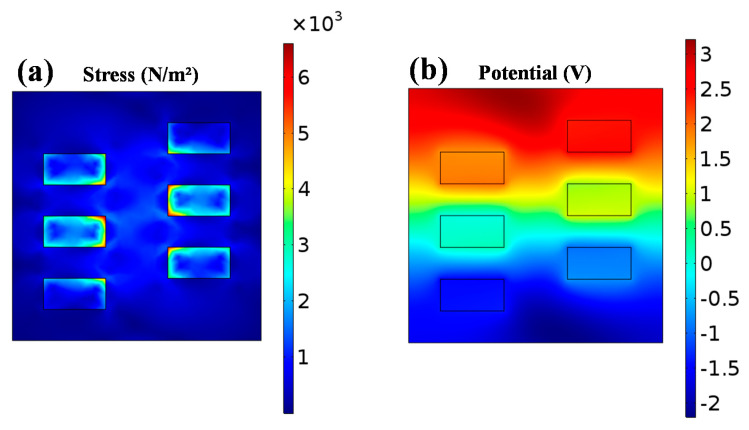
The simulated distribution of (**a**) stress and (**b**) the corresponding electric potential of 12.5 wt% BNdT NPs/PDMS f-PEH by COMSOL Multiphysics.

**Figure 7 micromachines-11-00966-f007:**
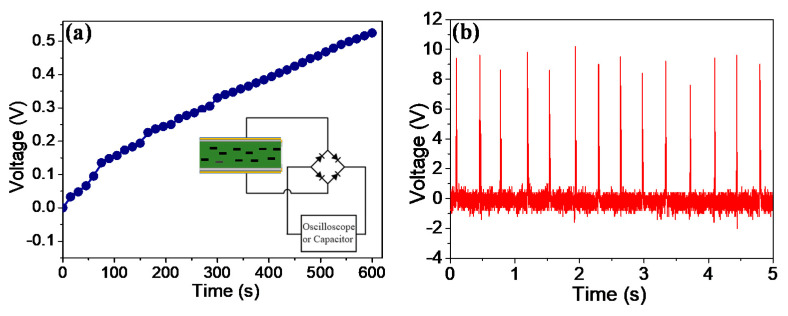
(**a**) The time dependent charging curve of a capacitor with a capacitance of 47 µF. Inset shows the schematic diagram of the circuit for rectification and energy storage; (**b**) the rectified output voltage *V* yielded from the 12.5 wt% BNdT NPs/PDMS f-PEH.

**Figure 8 micromachines-11-00966-f008:**
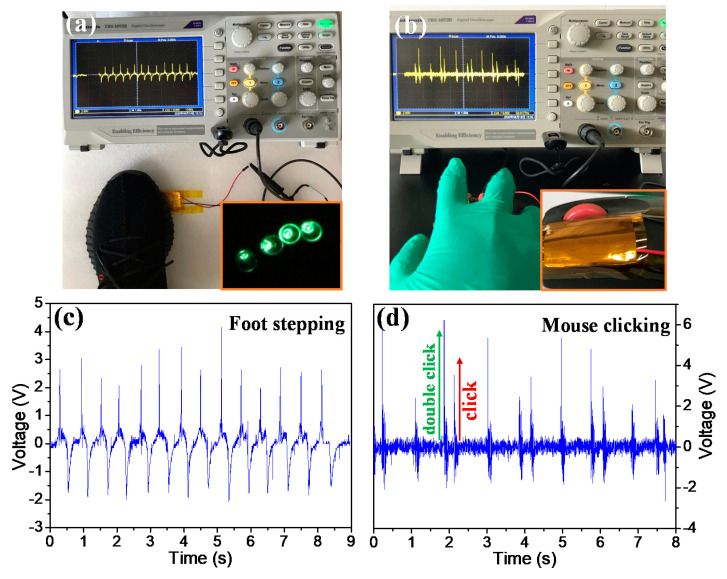
Applications in harvesting human motions of foot stepping (**a**) and (**c**) and mouse clicking (**b**) and (**d**) by the 12.5 wt% BNdT NPs/PDMS f-PEH. Inset in (**a**) is the photograph of lit commercial LEDs by the f-PEH by foot steeping or continuous tapping. Inset in (**b**) is the photograph of a f-PEH attached to the mouse.

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
