# Peer review of "Lead-Free Bi3.15Nd0.85Ti3O12 Nanoplates Filler-Elastomeric Polymer Composite Films for Flexible Piezoelectric Energy Harvesting"

_micromachines, 2020, doi:10.3390/mi11110966_

Round 1
Reviewer 1 Report
Dear authors, the topic of your paper is very interesting, it is very well written. To point to the relevance for possible applications, it would be helpful to add some results about long term stability of your device. Can a useage of 1000s of cycles during years be expected, or are improvements (encapsulation?) needed?
Reviewer 2 Report
The authors must describe the followings in detail,
- polarization method of the composite
- BNdT and PDMS properties(e.g. piezoelectric, elastic coefficients of the BNdT), and boundary conditions on FEM result
- type of vibration mode(e.g. d31, d33 etc.) of the harvester.
The authors also present opinions to the followings,
- I guess that more voltage will be generated on the harvester by foot stepping than mouse tapping. However the voltage generated looks similar in magnitude according to the figure (8).
- Correctness of time scale on figure (4), it looks like about 1.2Hz. And it is better to change vertical scale to μA.
- The power of 1.92μW was harvested by mouse clicking or foot stepping?
The authors must correct the misprints,
- …a power of 1.92mW …, to …a power of 1.92mW…on abstract,
- …under repetitive foot steeping…, to …under repetitive foot stepping…on the head of page 5.
